New insights into Mesozoic cycad evolution: an exploration of anatomically preserved Cycadaceae seeds from the Jurassic Oxford Clay biota

http://orcid.org/0000-0001-6590-405X Spencer Alan R.T. 1 alan.spencer@imperial.ac.uk
http://orcid.org/0000-0002-2803-9471 Garwood Russell J. 2 3
Rees Andrew R. 4
Raine Robert J. 5
Rothwell Gar W. 6 7
Hollingworth Neville T.J. 4 8
http://orcid.org/0000-0003-0286-8236 Hilton Jason 4 9
1 Department of Earth Science and Engineering, Imperial College London , London , UK
2 School of Earth and Environmental Sciences, University of Manchester , Manchester , UK
3 Department of Earth Sciences, The Natural History Museum London , London , UK
4 School of Geography, Earth and Environmental Sciences, University of Birmingham , Birmingham , UK
5 Geological Survey of Northern Ireland , Belfast , UK
6 Department of Botany and Plant Pathology, Oregon State University , Corvallis, OR , USA
7 Department of Environmental and Plant Biology, Ohio University , Athens, OH , USA
8 Science and Technology Facilities Council , Swindon , UK
9 Birmingham Institute of Forest Research, University of Birmingham , Birmingham , UK
Bateman Richard
Electronic publication date: 2017 Aug 28
Publication date: 2017
Volume: 5
Electronic Location ID: e3723
Received 2017 Jun 2; Accepted 2017 Jul 31
Copyright: © 2017 Spencer et al.
Copyright year: 2017
Copyright holder: Spencer et al.
License: This is an open access article distributed under the terms of the Creative Commons Attribution License, which permits unrestricted use, distribution, reproduction and adaptation in any medium and for any purpose provided that it is properly attributed. For attribution, the original author(s), title, publication source (PeerJ) and either DOI or URL of the article must be cited.
License URL: https://creativecommons.org/licenses/by/4.0/

Keywords: Palaeobotany, Fossil, Cycad, Cycadeospermum, Cycas, Oxford Clay Formation, Taphonomy, Gymnosperm, Ovule, 3D Reconstruction

Funding: NERC Award NE/E004369/1 Diamond Light Source experiment EE9244-1 This work was supported by NERC (Award NE/E004369/1) and the Diamond Light Source (experiment EE9244-1). The funders had no role in study design, data collection and analysis, decision to publish, or preparation of the manuscript.

==============================
Most knowledge concerning Mesozoic Era floras has come from compression fossils. This has been augmented in the last 20 years by rarer permineralized material showing cellular preservation. Here, we describe a new genus of anatomically preserved gymnosperm seed from the Callovian–Oxfordian (Jurassic) Oxford Clay Formation (UK), using a combination of traditional sectioning and synchrotron radiation X-ray micro-tomography (SRXMT). Oxfordiana motturii gen. et sp. nov. is large and bilaterally symmetrical. It has prominent external ribs, and has a three-layered integument comprising: a narrow outer layer of thick walled cells; a thick middle parenchymatous layer; and innermost a thin fleshy layer. The integument has a longitudinal interior groove and micropyle, enveloping a nucellus with a small pollen chamber. The large size, bilateral symmetry and integumentary groove demonstrate an affinity for the new species within the cycads. Moreover, the internal groove in extant taxa is an autapomorphy of the genus Cycas, where it facilitates seed germination. Based upon the unique seed germination mechanism shared with living species of the Cycadaceae, we conclude that O. motturii is a member of the stem-group lineage leading to Cycas after the Jurassic divergence of the Cycadaceae from other extant cycads. SRXMT—for the first time successfully applied to fossils already prepared as slides—reveals the distribution of different mineral phases within the fossil, and allows us to evaluate the taphonomy of Oxfordiana. An early pyrite phase replicates the external surfaces of individual cells, a later carbonate component infilling void spaces. The resulting taphonomic model suggests that the relatively small size of the fossils was key to their exceptional preservation, concentrating sulfate-reducing bacteria in a locally closed microenvironment and thus facilitating soft-tissue permineralization.

Introduction

Our present understanding of Jurassic vegetation is built largely upon the compression/impression fossil record. This provides important insights into plant morphology and diversity, as well as the temporal and spatial distributions of plants during the Mesozoic Era (Taylor, Taylor & Krings, 2009). This form of preservation is comparatively common in the geological record, and can include cellular details from cuticles that can be both informative systematically (Stockey, 1994) and environmentally (Barbacka & van Konijnenburg-van Cittert, 1998). Less common in the Mesozoic Era fossil record, however, is anatomical preservation in which plant cells and tissues are preserved in high fidelity, with permineralized fossil providing key insights into plant ontogeny, physiology, systematic affinities, and phylogenetic relationships that are not possible through other modes of preservation. Although the Jurassic flora of Britain is comparatively well-known from compression/impression assemblages and cuticles (van Konijnenburg-van Cittert & Morgans, 1999; Cleal et al., 2001; van Konijnenburg-van Cittert, 2008), only five British sites with anatomical preservation of Jurassic plant reproductive organs have been recognized previously (Fig. 1) amongst more frequent accounts of permineralized wood (Cleal et al., 2001; van Konijnenburg-van Cittert, 2008). The most important of these is the Aalenian Stage–Bajocian Stage Bearreraig Sandstone Fm at Bearreraig Bay on the Isle of Skye (Bateman & Morton, 1994; Bateman, Morton & Dower, 2000; Cleal et al., 2001; Dower, Bateman & Stevenson, 2004; Spencer et al., 2015) in which marine carbonate concretions preserve a range of vegetative and fertile organs produced by ferns, cycads and conifers. The same mode of preservation is found in the Kimmeridgian Kimmeridge Clay Fm at Eathie (Cleal et al., 2001), which contains exceptionally well preserved conifer cones (Seward & Bancroft, 1913; Rothwell et al., 2011, 2012, 2013) and Bennettitales (Seward, 1913). At Runswick Bay on the Yorkshire coast, Aalenian Stage in age, anatomically preserved plants—cycad leaves and cones and also cones of Bennettiales—occur in siderite concretions (Cleal et al., 2001). Other British Jurassic sites with permineralization preserve a single kind of plant organ. For example, carbonate-preserved pollen cones of cheirolepidiacean conifers are found in the Callovian Stage Kellaways Sand Mbr near Cirencester (Rothwell et al., 2007), and silicified ovulate cones of Cheirolepidiaceae from the Tithonian Purbeck Gp at Chicksgrove (Steart et al., 2014). Each of these assemblages is unusual because it preserves terrestrial plants in marine depositional environments where permineralization occurred rapidly after transport and deposition to prevent significant decay (Wignall & Pickering, 1993; Bateman, Morton & Dower, 2000).

Figure 1 UK lagerstätten locations and temporal ranges (Ma) in the Jurassic Period.

Time range chart showing depositional periods for the five UK lagerstätten.

Here, we describe the first anatomically preserved plant fossils from the Oxfordian Stage (late Middle Jurassic) Oxford Clay Fm of England (Fig. 2A). The fossils have been studied using a combination of traditional semi-destructive sectioning techniques (Spencer, Hilton & Sutton, 2013), X-ray microtomography (Spencer et al., 2015) and X-ray synchrotron microtomography (Steart et al., 2014). Through the recovery of excellent anatomical information from a few high-quality specimens, we establish a new genus and species of gymnosperm ovule Oxfordiana motturii ART Spencer & J Hilton gen. et sp. nov. Furthermore, because the ovules are preserved in pyrite and carbonate, which are readily differentiated using synchrotron radiation X-ray micro-tomography (SRXMT), we report in detail the taphonomy of fossil plants representative of those permineralized in marine depositional settings.

Figure 2 Stratigraphic placement and location map.

(A) stratigraphic summary showing the position of the ammonite Q. lamberti and Q. mariae biozones and their subzones, which correlate to the latest Callovian Stage and early Oxfordian Stage of the Jurassic, and the middle Oxford Clay Fm deposits; (B) map showing onshore exposures of the Oxford Clay Fm and the position of fossil recovery localities. Diagram based on Martill & Hudson (1991).

Materials and Methods

Two anatomically preserved specimens were collected from the Oxford Clay Fm at different sites. We consider these to be the same species due to consistent morphology and anatomical arrangement. Specimen BU 5265 was collected from Marston Maisey, near Cirencester (Fig. 2B; 51.667 N, −1.806 W; OS Grid Ref: SU 135 965) in 2011 by J. Motture, and specimen BU 5266 was collected from Dix Pit at Stanton Harcourt (Fig. 2B; 51.743 N, −1.420 W; OS Grid Ref: SP 412 049), in 2001 by RJR. Both permineralized specimens contain pyrite and calcium carbonate, and were found with pyrite decay and residual clay matrix attached to the outer surfaces. Both specimens are deposited at the Lapworth Museum of Geology, University of Birmingham.

Specimen BU 5265

The specimen has a black coalified outer surface with visible pyrite. It was initially photographed (Fig. 3) but later, due to pyrite decay, fragmented before further preparation occurred. The four largest pieces (numbered consecutively BU 5265.1–4) were subsequently mounted on metal stubs for analysis using scanning electron microscopy (SEM). SEM analyses proved inconclusive, so BU 5265.1 and BU 5265.2 were scanned using laboratory-based X-ray microtomography (XMT) on a Nikon Metrology X-Tek HMK-ST 225 scanner at the Natural History Museum (London, UK). Each fragment was scanned separately using a tungsten reflection target at 180–185 kV and 120 μA with a 0.25 mm copper filter, 0.354 s exposure time for 3,142 projections. The resulting 2,000 slice tomographic datasets (Datasets S1 and S2) had a voxel size of 16 μm and were visualized separately using previously described techniques (Spencer, Hilton & Sutton, 2013). Through the SPIERS software suite (http://spiers-software.org/; Sutton et al., 2012) we created three-dimensional (3D) isosurface-based false-color models (Models S1 and S2; Videos S1 and S2). The XMT results lacked fine detail, so the specimen was subsequently investigated with SRXMT on the I12 JEEP beamline at the Diamond Light Source (Oxfordshire, UK). The specimens were individually scanned using: a 53 keV incident beam; the “Module 2” optics, obtaining a 1.8× magnification; and with 1,800 projections collected using a PCO.4000 camera with an exposure of 0.45 s. The resulting tomographic datasets, consisting of 2,672 individual slices, have a voxel size of 5 μm (Video S3; Datasets S3–S5). The individual slice image data were multiplied, enhanced for brightness and contrast, and despeckled using ImageJ (Abràmoff, Magalhães & Ram, 2004). Volume rendering of the SRXMT datasets using the open-source volume renderer Drishti (http://sf.anu.edu.au/Vizlab/drishti/; Limaye, 2012) produced false-colored three-dimensional models that were visualized separately using previously described techniques (Hickman-Lewis et al., 2016); from which high-resolution images and virtual thin-sections were created (i.e. Figs. S1A–S1E).

Figure 3 Photographs of specimen BU 5265 prior to fracturing from pyritic decay.

(A) Side of ovule, with pointed apex and flattened base, showing longitudinal ribs which merge toward the apex; (B) side view rotated 90° side from A, showing “bored” hole (arrow), and bilateral symmetry of the ribs and base; (C) basal view showing 12 ribs and the attachment scar; (D) oblique apex view showing the ribs tapering to a point, “bored” hole indicated by arrow. Scale bar = 10 mm.

Specimen BU 5266

The specimen was photographed and measured, during which, however, the exterior surface started to crumble due to extensive pyrite decay (Fig. 4). To stabilize the specimen, it was embedded in low viscosity resin (Buhler Epo-Thin®) under vacuum, drawing resin into the fossil and excluding contact between air and pyrite. After curing for 24 h, samples were re-embedded in blocks of epoxy resin, allowing them to be orientated and mounted for precision cutting to reveal anatomical information. This used a low-speed Buhler Isomet wafering saw with a 0.1 mm wide diamond edged cutting blade as outlined by Hass & Rowe (1999).

Figure 4 Photographs of specimen BU 5266 prior to serial wafering and peeling.

(A) Side view of ovule showing decayed outer surface with eroded ribs; (B) view of heavily decayed apex showing the rounded 12 ribbed nature of the ovule; (C) oblique side view showing ribbed outer surface, with zones of decay exposing inner integument, red staining on the base shows attachment scar; (D) basal view, note the four relatively intact ribs seen toward the image top, and the attachment scar denoted by the red staining. Scale bar = 10 mm.

The specimen was initially halved longitudinally near to the middle of the ovule. One half was then prepared in longitudinal section and the other in transverse section. Cut surfaces were prepared using the acetate peel technique (Galtier & Phillips, 1999) through 10 s grinding with 400 carborundum to prepare the surfaces, and 5% HCL to etch the carbonate for 30–40 s. Peels were mounted using Eukitt. After surfaces were peeled, the specimen was further sectioned to reveal longitudinal/radial and transverse planes, two peels being created from each cut surface. Finally, each block was ground and mounted on glass slides for microscope observation using reflected incident light. The position of peels and wafered sections are shown in Fig. 5.

Figure 5 Diagram showing position of cuts, wafers and peels taken from specimen BU 5266.

Red lines, saw cuts; p, peel; w, wafer.

Selected slide mounted wafers were then studied using the I12 JEEP beamline at the Diamond Light Source (a technology that was unavailable when the specimen was sectioned). All slides were scanned by collecting 1,800 projections using “Module 2” optics: BU 5266.37, BU 5266.38, BU 5266.39 at 70 keV with an exposure of 0.5 s, and BU 5266.1, BU 5266.3, and BU 5266.30 at 53 keV with an exposure of 0.45 s. The resulting 2,672 slice tomographic datasets have a voxel size of 5 μm (Datasets S6–S10). We visualized data using Drishti to produce greyscale three-dimensional models from which high-resolution images and virtual thin-sections were created (e.g. Figs. S1F–S1J). We note that the high brightness of the X-ray beam caused dark brown discoloration of the glass slides.

Figures and videos

Images were processed (cropped, rotated, edge enhanced and equalized for hue and brightness) in GIMP 2, ImageJ (Abràmoff, Magalhães & Ram, 2004) and Corel® Paint Shop Pro® Photo X2. Figures were constructed in CoralDraw® X7 and Inkscape. Blender™ (Garwood & Dunlop, 2014) was used to raytrace images from the 3D models for both figures and video animations. The latter were edited within Blender™ (Videos S1–S3).

Geological information

Both specimens were found in the Oxford Clay Fm (Martill & Hudson, 1991): a 17 m-thick primarily argillaceous unit with a diverse and distinctive macrofauna (Martill & Hudson, 1991). Specimen BU 5265 was found in a silty blue gray clay lens. In association were thin shell beds co-occurring with the ammonites Creniceras sp. (Munier-Chalmas), Taramelliceras sp. (Campana) and Quenstedtoceras mariae (d’Orbigny). The last of these species defines the earliest Oxfordian Stage Q. mariae Biozone, found at the base of the Weymouth Mbr of the Oxford Clay Fm (Fig. 2A). Ammonites associated with Dix Pit specimen BU 5266 include Distichoceras bicostatum (Stahl), Kosmoceras spinosum (Sowerby), Alligaticeras sp. and Quenstedtoceras lamberti (Sowerby). The latter—present in large numbers—is diagnostic of the Q. lamberti ammonite subzone of the Q. lamberti Zone (Martill & Hudson, 1991). Furthermore, this ovule was collected a few meters below the Lamberti Limestone that represents the top of the Stewartby Mbr (Fig. 2A) of the Oxford Clay Fm, placing this in the latest Callovian Stage. Hence, ammonite biostratigraphy places both the fossils spanning the boundary of the Callovian Stage (Middle Jurassic) and Oxfordian Stage (Upper Jurassic), at about 163.5 Ma (Fig. 2A).

The micro- and macro-fauna, coupled with the sedimentology of the Oxford Clay Fm, demonstrate that it was deposited under shallow marine conditions in an epicontinental sea (Anderton et al., 1979; Martill & Hudson, 1991; Cope, Ingham & Rawson, 1992). Plant material found within the formation was transported into a marine depositional setting from surrounding landmasses of which the nearest was the London-Brabant Massif to the southeast (Cope, Ingham & Rawson, 1992). Other permineralized fossil plants are known from the Oxford Clay Fm but comprise woody conifer trunks and branches (Porter, 1863; J. Hilton & ART Spencer, 2006–2017, unpublished data), and another kind of gymnosperm ovule (A. R. Rees et al., 2016, unpublished data). These specimens are also preserved in pyrite and calcium carbonate, suggesting that the mode of plant permineralization outlined here may be widespread.

Results

Specimen BU 5265 (Fig. 3) is more complete and better preserved than specimen BU 5266 (Fig. 4) because the latter has suffered from diagenetic replacement of organic tissues by cubic pyrite during fossilization. Coupled with extensive pyrite decay, this process has removed the external layers—cellular details only remain in a few parts of the ovule. By contrast, BU 5265 shows few signs of pre-burial taphonomic decomposition or damage. It is externally complete, with the specimen partially encased in clay from its depositional environment. The following description is based primarily on BU 5265, augmented by additional information from BU 5266 where available.

Gross morphology

The ovules are 16.8–21.6 mm long and 17.9–21.9 mm in maximum diameter (1/3 of the ovule length from base), and comprises a ribbed integument with a distal micropyle surrounding a central nucellus (Figs. 3, 6 and 7). The integument is ovate–orbicular in longitudinal section with a bluntly rounded chalaza, tapering to a pointed apex (Figs. 3, 6A–6C, 7C, 7D, 8A and 9A–9C). It is circular in cross-section with prominent ribs (Figs. 3C, 3D, 6D–6F, 7A, 7B, 8 and 9G–9I). The chalaza is flattened and 6.0–6.7 × 7.5–11.3 mm wide, with a 4.3–4.6 × 6.6–7.2 mm concave base that marks the position of attachment to the parent plant (Figs. 3C, 4A, 4C and 4D). The three-layered integument lacks an external epidermis, even where the fossils are encased in the clay matrix. This observation suggests it was absent at the time of fossilization, and that the ovule had undergone some prior decay. The outer integumental layer is thin, and comprises radially aligned thick-walled cells—we consider this to be an outer sarcotesta (Figs. 6, 7 and 9D–9I). By contrast, the thicker middle layer consists of thin walled parenchyma cells, which we propose is an inner sarcotesta (Figs. 6, 7, 9D–9I, 10A and 10B). The innermost integumentary layer contains thin-walled parenchyma cells organized in two to five rows. Individual cells are elongate here, and aligned parallel to the ovule circumference—we consider this layer to be the endotesta (Figs. 6, 7, 9D–9F, 9C and 10B). A longitudinal interior integumentary groove, approximately 1.4 mm wide and more pronounced at the apex and bisecting the micropyle region, aligns with two of the integumentary ribs (Fig. 11). The nucellus is free from the integument but partially collapsed in BU 5265 (Figs. 6A–6C). It is seen to follow the integumentary margin in BU 5266 (Figs. 7, 10A and 10B). The nucellus envelops a single functional megaspore and has an apical nucellar beak (Figs. 6, 7C and 7D). No vascular tissues have been identified in either seed. Between two of the ribs along the midpoint of BU 5265 (arrowed on Figs. 3B and 3D), a rounded but irregular hole approximately 1 mm in diameter penetrates into the specimen’s interior. We interpret this feature as early post-mortem boring by an organism that cut through all of the seed layers prior to permineralization.

Figure 6 Virtual 25 μm thin-sections showing gross morphology and anatomy of BU 5265.1 created from the Drishti volume rendered three-dimensional model.

Each figure shows (A, D) full SRXMT data in grayscale with enhanced brightness and contrast; (B, E) false-color image showing only the high density pyritic material; and (C, F) the medium density organic and carbonate-rich materials. (A–C) longitudinal section through ovule. Showing at the apex a nucellar beak (Nb) that protrudes from the top of the nucellar membrane (N) within a defined micropyle (Mi) region. Several major layers of the integument (Into = outer; Intm = middle) are seen; (D–F) Cross-section across ovule, approximately 1/4 distance from base to apex, line of section as marked on (A–C). All scale bars = 1 mm.

Figure 7 Gross morphology and internal feature distribution of BU 5266 viewed as wafers and virtual thick-sections.

Wafers (A, C) viewed in reflected light photographed using a Zeiss Tessovar, and virtual thick-sections (B, C) created from the SRXMT data in the Drishti volume rendered. (A, B) cross‐sectional wafer near ovule base with its three identified layers (Into = outer; Intm = middle; Inti = inner), nucellus (N), and megagametophyte (Mg); (B) longitudinal wafer near ovule center with its three identified layers (Into = outer; Intm = middle; Inti = inner), nucellus (N), megaspore membrane (Mm), and megagametophyte (Mg). Slides: A, B = BU 5266.37; C, D = BU 5266.3. All scale bars = 2 mm.

Figure 8 Three-dimensional isosurface-based false-colored reconstruction model, BU 5265.1.

(A, B) and BU 5265.2 (C–D), showing gross morphology of the ovule. (A) longitudinal view showing ovule with partly remaining integument. The ribbed outer integument (Into) can be seen, with furrows containing Oxford Clay (Cl), overlying the middle and inner integumentary layers (IntM+I). The cavity formed by the integument is digitally filled (Int. Cav.). Note the micropyle protruding from the apex (Mi); (B) basal view showing the heavily ribbed nature of the integument with attached Oxford Clay; (C) longitudinal view of a basal integument section showing the ribs terminating at toward the attachment scar; (D) longitudinal view 90° from (C) showing the flattened nature of the base and multi-layered construction of the integument; (E) view of the integument section from the apex showing the middle and inner integument packing the central portions of the ribs. Scale bar = 10 mm. Animations of these reconstructions can be seen in (Videos S1 and S2).

Figure 9 Virtual 25 μm thin-sections showing gross morphology and anatomy of BU 5265.1 created from the Drishti volume rendered three-dimensional model.

(A, D, G) full SRMT data in grayscale with enhanced brightness and contrast; (B, E, H) false-color image showing only the high density pyritic material; and (C, F, I) the medium density organic and carbonate rich materials. (A–C) enlarged longitudinal view of the apex seen in Figs. 6A–6C. Showing a nucellar beak (Nb) that protrudes from the top of a possible small pollen chamber (Pc), the nucellar membrane (N), and the micropyle (Mi) region; (C–F) longitudinal section of integument, through the maximum width of a rib, line of section as marked on Figs. 6D–6F. Showing the three identified layers of the integument (Into = outer; Intm = middle; Inti = inner), F also shows the location of the nucellus (N); (G–I) enlarged cross-sectional view of integument, as marked on Figs. 6D–6F. Showing two ribs, with Oxford Clay infill in the rib furrow, and the integument with its three identified layers (Into = outer; Intm = middle; Inti = inner). All scale bars = 1 mm.

Figure 10 Physical wafer and peel sections of BU 5266 showing state of preservation and anatomy.

Wafers (reflective light) and peels (transmitted) light were photographed using a Zeiss Tessovar. (A) longitudinal wafer showing the extensive pyrite and pyritic decay of the integument with patches of preserved middle integument (Int) and small portions of nucellus (N). Slide BU 5266.3. (B) longitudinal peel showing integument with its thin inner layers (Inti) and middle layer (Intm). The external surface of the integument is degraded (left of image). Slide BU 5266.12. (C) enlarged view from A showing the preserved middle integument cells. Slide BU 5266.3 (D) longitudinal wafer showing micropyle (Mi) at the middle top of the ovule, the thin nucellus (N) closely following integument, and the megaspore membrane (Mm) and megagametophyte (Mg) within the integumentary cavity. Slide BU 5266.3. (E, F) high magnification of wafer showing the construction of the nucellus (N). Slide BU 5266.2. (G) high magnification of preserved cells from within the megaspore membrane interpreted as megagametophyte tissue (Mg). Slide BU 5266.37. Scale bars: A, D = 1 mm, E = 0.5 mm, all others 0.25 mm.

Figure 11 Three-dimensional isosurface-based false-colored reconstruction model of BU 5265.1 showing a digitally filled integumentary cavity cast.

(A) longitudinal view looking along the central apex infilled internal integumentary groove; (B) apex view of the infilled internal integumentary groove with the micropyle protruding (Mi); (C) longitudinal view showing the infilled internal integumentary groove with central micropyle in profile. Scale bars = 5 mm.

Integument

The integument of both specimens has 12 prominent ribs, each 2.44–4.5 mm (x¯ 3.39 mm; n = 6) wide and 0.6–0.9 mm (x¯ 0.75 mm; n = 6) high (Figs. 3 and 4). These start at the base of the ovule and some extend for the entire length; residual sediment is retained in some furrows between adjacent ribs (Figs. 3 and 8; Models S1 and S2; Videos S2 and S3). In the bottom 1/3 of the ovule, ribs are narrower and shorter than they are apically (Fig. 3C), where the ribs merge together and become less pronounced (Figs. 3A, 3B and 4D). Both ribs and furrows are rounded in profile (Figs. 3C, 8B and 8E; Videos S2 and S3). At the widest point of the ovule, the integument is ∼1.2–2.5 mm thick in the furrows, increasing to ∼2.7–3.5 mm in the center of ribs (Figs. 3 and 8).

The outermost integument—that of the outer sarcotesta—is thin, typically 0.17–0.46 mm (Figs. 6, 7A, 7B and 9D–9I). It consists of elongate, thick-walled cells aligned perpendicular to the ovule body, forming a fibrous palisade (Figs. 9D–9I). Individual cells here are ∼52–72 μm long and ∼16–32 μm wide.

The middle and thickest integumentary layer—that of the inner sarcotesta—is readily distinguished in the furrows as its cells are orientated perpendicular to the outer sarcotesta (Figs. 9G–9I). On the ribs the inner and outer sarcotesta boundary is less distinct as the cells of both layers are parallel (Figs. 9D–9I). The inner sarcotesta contains large, isodiametric parenchyma cells, 50–68 μm in diameter and with cell walls 8–16 μm thick (Figs. 9G–9I and 10A–10C). These form the bulk of the integument; this layer is 0.8–1.5 mm thick at furrows, increasing to 2.4–2.8 mm at the rib centers (Figs. 9D–9I). Cell orientation in the inner sarcotesta is symmetrical along the rib midline where they are aligned toward the outside edge of the rib, whereas in the furrows the cells are organized perpendicular to the rib orientation (Figs. 9D–9I and 10B). The central portion of each rib has the largest cells; size decreases and shape becomes more elongated toward the outer and inner margins (Figs. 9G–9I).

Endotestal cells are typically poorly preserved. Where complete, the layer is 50–180 μm thick and comprises two to five rows of small, thin-walled parenchyma cells that are elongated parallel to the ovule circumference (Figs. 9D–9I and 10B). These line the integumentary cavity, and individual cells range from 47–91 μm (x¯ = 71 μm; n = 20) in length, and 16–36 μm (x¯ = 27 μm; n = 20) in width. Cell walls are 8–14 μm (x¯ = 11 μm; n = 40) thick (Figs. 9D–9I). The anatomy at the internal groove is poorly preserved at the apex, and basally is physically missing (Figs. 11; Figs. S1A and S1B). The interior morphology of the integument shows a longitudinal groove that bisects the micropyle region, and is 1.35–1.5 mm wide at this point (Fig. 11). The groove measures ∼7.0 mm in length and ∼1.0 mm in height (Fig. 11). This feature becomes less pronounced as it descends from the apex, terminating toward the ovule base. Lateral compression of the seed has distorted the lower integument making determination of the indent in the lower ovule regions difficult (Figs. 11; Figs. S1A, S1B and S1E). The micropyle is 0.8 mm in diameter (Figs. 12B–12F).

Figure 12 Three-dimensional isosurface-based false-colored reconstruction model of BU 5265.1 showing internal ovule morphology.

(A) ovule, with Oxford Clay and digitally filled integumentary cavity removed, showing the multi-layered integument (Into and IntM+I) surrounding a large nucellus (N) with micropyle channel at apex (digitally infilled; Mi); (B) longitudinal section through the middle of A. The nucellus, which has deteriorated toward the back of the view, possesses a small nucellar beak at is apex (Nb), that protrudes into the micropyle channel (digitally infilled; Mi); (C) Nucellus (N) and digitally infilled micropyle (Mi), shown without surrounding integument; (D) Enlarged apex view of (C), with micropyle channel (Mi) shown as semi-transparent, and nucellus beak (Nb) within. Dashed line indicates the boundary between the digitally infilled micropyle and the nucellar beak; (E) longitudinal section through the apex of (D). Showing the nucellus (N), with small pollen chamber (Pc) that resided below the nucellar beak (Nb). The digitally infilled micropyle (Mi) is shown; (F) Cross-section through (E), showing relationship between nucellus (N), the nucellar beak (Nb), and the micropyle (Mi). Scale bars: A, B = 5 mm; C = 1 cm; D–F = 1 mm.

Nucellus and megaspore

Where preserved, the nucellus—including the nucellar apex and the integumentary micropyle—is heavily encrusted and infilled with pyrite (Figs. 9A–9C and 10D). This leaves only a faint residue of coalified material, visible under XMT, marking the position of the cellular membranes (Figs. 7B, 7C and 9A–9F). Individual cells are not preserved (Figs. 6A–6F and 9A–9C; Video S1).

The nucellus is 10.7–13.9 mm long (Figs. 6A–6C and 12C) and a maximum of 8.2–8.4 mm wide (Figs. 6A–6C and 10C). It is unicellular and 18–49 μm thick, its cells averaging 19 × 33 μm (Figs. 10E and 10F). The structure closely follows the integument but has collapsed and shrunk in places (Figs. 10A, 10B, 10D, 11A and 11B). Pyrite growth enveloping the nucellar membrane makes the original position of the nucellus uncertain (Figs. 6, 10A and 10E). However, we note it is adnate to the base of the integument in both specimens (Figs. 6 and 7) and is in very close association with the integumentary wall throughout BU 5266 (Figs. 7, 10A, 10B and 10E) except toward the apex (Fig. 10D). The nucellar apex is situated directly below, and protrudes into the integumentary micropyle (Figs. 6A–6C, 9A–9C and 12B–12F). It comprises a small pollen chamber and distally a conical nucellar beak (Figs. 6A–6C, 9A–9C and 12B–12F) that is 0.74 mm high, and 0.42 mm wide at it base, narrowing to an apex 0.14 mm wide. The wall of the nucellar beak is 25–35 μm thick, and the floor in contact with the underlying nucellus is ∼20 μm thick (Figs. 9A–9C and 12E).

Within the nucellus of BU 5266 is a megaspore membrane (Fig. 10F), preserved by (and largely filled with) micro-crystalline pyrite (Fig. 7A and 7C). The membrane of BU 5266 is ∼5 μm thick, within which a small zone of preserved cells ∼32 μm wide is interpreted as gametophytic tissue (Fig. 10G). Archaeogonial tissues have not been identified.

Systematic Palaeobotany

Superdivision SPERMATOPHYA

Division CYCADOPHYTA Bessey (1907)

Class CYCADOPSIDA Brongniart (1843)

Order CYCADALES Pers. ex. von Berchtold & Presl (1820)

Family CYCADACEAE Persoon (1807)

Oxfordiana motturii gen. et sp. nov. ART Spencer & J Hilton

Generic Diagnosis: Anatomically preserved ovule with bilateral symmetry, ovate–orbicular in longitudinal section, circular in external cross-section, with pronounced longitudinal ribs and internal integumentary groove dividing the seed coat into two valves. Ribs rise from a flattened, concave base, with central attachment scar, extending to a pointed apex with conical micropyle. Integument three layered. Nucellus attached to integument at base; nucellar apex comprising a small pollen chamber and nucellar beak.

Specific Diagnosis: Twelve ribs rounded in profile, narrower in basal 1/3 of ovule with adjacent ribs merging together. Less pronounced in apical 1/3 of ovule. Integument thickest at center of ribs. Thinning laterally and thinnest in furrows between ribs. Outer integument forming thin, fibrous palisade of elongate, thick-walled cells aligned perpendicular to ovule body. Middle integument thickest, comprising variable-sized isodiametric parenchyma cells aligned perpendicular to outer integument in rib furrows, becoming parallel within ribs, thus symmetrical along the rib mid-line. Inner integumentary layer represented by two to five rows of small, thin-walled, elongate parenchyma cells aligned with ovule circumference. Integument with conical micropyle. Pollen chamber with small nucellar beak that projects into micropyle.

Localities: Dix Pit near Stanton Harcourt (51.743 N, −1.420 W; OS Grid Ref: SP 412 049) and Marston Maisey, near Cirencester (51.667 N, −1.806 W; OS Grid Ref: SU 135 965), UK.

Horizon: Uppermost Stewartby Mbr to lowermost Weymouth Mbr of the Oxford Clay Fm.

Stratigraphy and age: Quenstedtocdras lambertii ammonite subzone of the Quenstedtocdras lambertii Biozone to Cardioceras mariae Biozone; latest Callovian Stage (Middle Jurassic) to earliest Oxfordian Stage (Upper Jurassic).

Holotype: BU 5265 consisting of 4 parts (BU 5265.1–4).

Paratype: BU 5266.

Etymology: The generic epithet is derived from the Oxford Clay Fm from which the species is recognized. The specific epithet is named in honor of Julian Motture, who collected the type specimen.

Depository: Lapworth Museum of Geology, University of Birmingham.

Discussion

Comparison with other gymnosperm ovules and evolutionary implications

Oxfordiana motturii gen. et sp. nov. is a large, bilaterally symmetrical, ribbed ovule that has similarities in terms of its gross morphology with seeds produced by several taxonomic groups. It is closest in morphology to the Jurassic ovule Cycadeospermum Saporta (Saporta, 1875) from the Oxfordian Stage of France. The diagnosis of Saporta includes ovate ovules, some of which are ovate–oblong, possess longitudinal ribs, have a rounded base with an attachment scar, and have a tapered apex (e.g. Fig. 13). However, Saporta also indicated that the longitudinal ribs may in fact be angular vegetative leaflets forming a cupule around the ovule (Saporta, 1875, p. 237); an interpretation we consider unsupported speculation. Cycadeospermum schlumbergeri Saporta (Saporta, 1875; Plate CXVII, Figs. 11 and 12) is incompletely characterized and has not been described in detail. Nevertheless, Saporta’s description and his drawings of the species demonstrate it to have an external morphology that is similar to the species documented here (Figs. 13A and 13B). Cycadeospermum schlumbergeri is ovate–conical, with a truncated base with an attachment scar. From the four-keeled base, ribs form that extend to the ovule’s apex (Saporta, 1875). Ovules of this species are 1.6–2.4 cm long, and 1.3–1.9 cm wide, and hence are of similar shape and size to the British specimens described here. Anatomical details are not known in the French specimens, preventing more detailed comparison. Unfortunately, the C. schlumbergeri specimens on which Saporta based his account belonged to private collections and do not appear to have been deposited in a museum to allow future comparisons. We can at least state that Cycadeospermum schlumbergeri comes from a coeval depositional unit, adjacent to the shallow marine Channel Basin, and is potentially derived from the Armorica Massive. Without access to the type specimen of the French species for restudy, and using only the rudimentary illustrations and non-histological descriptions available from Saporta’s work, we consider it preferable to describe the fossils presented here as a new genus. Saporta’s concept of Cycadeospermum is based entirely on gross external morphology with no histological preservation known, thereby differentiating it sufficiently from Oxfordiana erected to contain the Oxford Clay Fm ovules possessing anatomical preservation.

Figure 13 Original pen and ink drawings of Cycadeospermum ovules from Saporta (1875, 1891).

(A) C. schlumbergeri from Plate CXVII Fig. 11 of Saporta (1875) showing ovule looking down on apex with ribbed integument and lateral opening (arrow); (B) C. schlumbergeri from Plate CXVII Fig. 12 of Saporta (1875) showing longitudinal view of ovule with ribbed integument, pointed apex and flattened base; (C) C. berlieri from Plate CCXCVIII Fig. 3 of Saporta (1891) showing longitudinal view of ovule with ribbed integument; (D) Smaller specimen of C. berlieri from Plate CCXCVIII Fig. 4 of Saporta (1891) showing longitudinal view of ovule with ribbed integument; (E) C. choffati from Plate CCXCVIII Fig. 5 of Saporta (1891) showing longitudinal view of ovule with ribbed integument, with attachment point at base. Scale bar = ca. 1cm (based on information given by Saporta in text).

The ovules described here bear a general resemblance to those of Cycadales, a group that can be traced back to the Palaeozoic Era, reached peak diversity during the Jurassic and Cretaceous periods, apparently underwent a more recent evolutionary radiation (Nagalingum et al., 2011), and includes extant members. Of the extant genera, Bowenia is sometimes classed as a “living fossil” as it can be traced back into the Mesozoic Era (Nagalingum et al., 2011). All cycad seeds present a unique combination of features among seed plants (Brenner, Stevenson & Twigg, 2003) and in extant plants display two distinct types of ovule morphology (Stevenson, 1990). Platyspermic ovules, restricted to Cycas, among living cycads, are oval to distinctly flattened, bi-lobed, with an apical depression, whereas all other extant cycads have radiospermic ovules. Ovules of Cycadeocarpus from the Jurassic of Haida Guai (previously known as Queen Charlotte Islands), Canada described by Dawson (1872) and subsequently re-investigated by Chaloner & Hemsley (1992), are at 4–5.25 × 4.25–4.5 cm substantially larger than O. motturii. They possess a series of ∼18 canals within the integumentary tissue surrounding the micropyle (i.e. coronula), a feature absent from the Oxford Clay Fm specimens, the genus Cycas, as well as a currently unnamed species of cycad seeds from the same locality as Cycadeocarpus (Rothwell, Stockey & Stevenson, 2017). The bilateral symmetry (more specifically, 180° rotational symmetry of Rothwell, 1986) is intimately associated with the germination mechanism of Cycas and probably also with the anatomically preserved Jurassic fossil seeds. There are, however, notable morphological differences between O. motturri and all other Mesozoic Era cycads. Ovules of the Jurassic megasporophyll Beania Carruthers are nearly circular, lack prominent longitudinal ribbing, and are smaller than these Oxford Clay Fm ovules. For example, Beania gracilis is ∼16 mm long and ∼13 mm wide, and Beania mamayi is ∼4 mm wide (Harris, 1964). Furthermore, although Beania species are known only from compression/impression fossils, precluding full comparisons, B. mamayi has resin bodies in the integument, and both B. gracilis and B. mamayi have cuticles that demonstrate the presence of a stony layer (Harris, 1964), features that are absent from the Oxford Clay Fm ovules. In terms of comparison with extant, broadly similar cycads, ovules of Dioon (Chamberlain, 1906), Bowenia (Kershaw, 1912) and Cycas (Stopes, 1905; Seward, 1917) have an integument with a uniseriate epidermis, an outer parenchymatous sarcotesta, a stony sclerotesta, and innermost fleshy endotesta. Additionally, in most cycads the endotesta has a similar thickness to the stony sclerotesta, and does not appear to collapse at maturity as in most other gymnosperm ovules (Seward, 1917; Taylor, Taylor & Krings, 2009). However, these genera possess a twin vascular system with bundles in the sarcotesta and endotesta extending toward the micropyle (Kershaw, 1912; Coulter & Chamberlain, 1938). This organization differs from Oxfordiana, where vascular bundles have not been found in the sarcotesta and endotesta. We note that Dioon has an abscission layer (Chamberlain, 1906) where it sheds from the sporophyll, which results in a chalazal scar similar to the basal feature in Oxfordiana.

Cycas is unique among the extant cycads in having a novel germination mechanism in which the seed coat splits into two equal valves at the micropylar end to allow germination (Stevenson, 1990). This appears to be comparable with the integumentary groove in the apex of Oxfordiana, and also with the valved integument of the unnamed seed from Haida Gwai (Rothwell, Stockey & Stevenson, 2017) which we consider also to have facilitated germination. These characters appear to represent a common germination mechanism, found nowhere else outside the Cycadeaceae among living and fossil seed plants. If this germination mechanism represents a synapomorphy of the Cycadaceae, then the novel integumentary histologies and vascular systems of Oxfordiana and the unnamed seed from Haida Gwai suggest that the fossil taxa represent stem group representatives of the Cycadaceae.

Oxfordiana motturii also superficially resembles the ovules of Ginkgoales. These are readily identified through the presence of a dense sclerotesta with two to three ribs, and a ribless fleshy sarcotesta (Seward & Gowan, 1900). Czeckanowskialeans are a group reputedly closely allied with Ginkgoales (Taylor, Taylor & Krings, 2009). Their ovules, assigned to Leptostrobus Heer, are borne in ∼5 mm-long seed-capsules, each of which contains up to five ovules. These are significantly smaller than the specimens presented here (Heer, 1876; Harris, Millington & Miller, 1974; Liu, Li & Wang, 2006). Anatomical features of czecknowskialeans are unknown, preventing further comparison.

Pteridosperms were present during the Mesozoic Era, but their ovules were much smaller than the Oxford Clay Fm specimens. For instance, ovules of the Caytoniales (Reymanówna, 1973) and Pentoxylales (Sahni, 1948; Bose, Pal & Harris, 1985; Drinnan & Chambers, 1985; Césari et al., 1998) are ∼2 mm long. Those of the pentoxylalean genus Carnoconites Srivastava have a thick fleshy sarcotesta, and are have been described as both platyspermic with two sclerotestal ribs (Srivastava, 1935, 1937, 1946; Sahni, 1948; Bose, Pal & Harris, 1985; Taylor, Taylor & Krings, 2009) and radiospermic with no ribs (Sharma, 2001). Some mature bennettitalean ovules (e.g. Cycadeoidea Buckland, Monanthesia Wieland ex Delevoryas) are similar in size to those of the Bennettitales and possess an elongate distal extension of the integument forming the micropylar tube. Furthermore, they lack a pollen chamber, instead having a solid nucellar plug (Stockey & Rothwell, 2003). In Williamsoniaceae (Williamsonia Carruthers) ovules are larger—up to 8 mm long—and circular, have an elongate micropylar tube, and nucellar characters comparable with those of Cycadeoidaceae (Sharma, 1970, 1980; Stockey & Rothwell, 2003).

The ovules of Triassic Petriellales are triangular and significantly smaller (∼1 mm diameter) than the Oxford Clay Fm ovules (Taylor, Fueyo & Taylor, 1994). Corystospermales, also most common in the Triassic, differ from these specimens as they are considerably smaller (e.g. seeds of Umkomasia macleanii H.H. Thomas are 5 mm long), bilaterally symmetrical, and possess a single integumentary layer comprising small thin-walled isodiametric cells. In Umkomasia resinosa Klavins, Taylor & Taylor the integument also contains secretory cavities (Klavins, Taylor & Taylor, 2002). Members of the group also possess a slightly curved bifid micropylar extension (Axsmith et al., 2000; Klavins, Taylor & Taylor, 2002).

The ovules described here also differ from gnetalean plants. All extant (Ephedra L., Gnetum L., and Welwitschia Hooker) and most extinct gnetalean genera, have chlamydospermous ovules, with one or more distinctive seed envelopes or bracteloles enclosing the ovule (Taylor, Taylor & Krings, 2009; Friis, Pedersen & Crane, 2014). They also possess an extended micropylar tube protruding above the seed envelope or bracteoles (Friis, Pedersen & Crane, 2013). We note that ovules of the extinct species Protoephedrites eamesii Rothwell & Stockey (Rothwell & Stockey, 2013) lack an outer seed envelope and have a diminutive micropylar tube but are significantly smaller than the Oxford Clay Fm ovules. In general, gnetalean ovules possess thinner membranous and relatively undifferentiated integuments than those presented here (Rothwell & Stockey, 2013; Friis, Pedersen & Crane, 2014).

We have also compared the Oxford Clay Fm ovules with older species. They are superficially similar to the ovules of Palaeozoic Era medullosan pteridosperms, in sharing large size and similar nucellar apex structure (Taylor, Taylor & Krings, 2009; Spencer et al., 2013). Nevertheless, medullosan ovules have a prominent sclerotesta (stony layer) that is absent from the Oxford Clay Fm ovules. In addition to having radial, rather than bilateral symmetry, all of the eight well-characterized genera of anatomically preserved medullosan ovules within the traditional trigonocarpalean group (Seward, 1917) differ from the Oxford Clay Fm specimens in being radially symmetrical and in possessing a ribbed sclerotesta (Spencer et al., 2013; Spencer, Hilton & Sutton, 2013). Furthermore, Polylophospermum Brongniart, Polypterospermum Brongniart (Brongniart, 1874; Combourieu & Galtier, 1985), Hexapterospermum Brongniart (Brongniart, 1874; Taylor, 1966) and Hexaloba Dunn, Mapes & Rothwell (Dunn, Mapes & Rothwell, 2002) have six-lobes in transverse section, whereas Pachytesta Brongniart and Stephanospermum Brongniart have three prominent ribs (Brongniart, 1874). Codonospermum Brongniart is distinguished by a two-chambered integumentary cavity with the nucellus and associated chamber above an empty lower chamber (Brongniart, 1874; Seward, 1917; Combourieu & Galtier, 1985). Finally, Rhynchosperma Taylor & Eggert differs in its possession of apical integumentary appendages and bulbous hollow lobes as part of a stellate micropylar canal (Taylor & Eggert, 1967; Dunn, Rothwell & Mapes, 2002). We also note that medullosan pteridosperms are only known from the Carboniferous and Permian periods (Phillips, 1980; Hilton & Cleal, 2007), so are considerably older than the Oxford Clay Fm ovules.

Based on these comparisons, it is clear that O. motturii is externally morphologically similar, to the coeval Cycadeospermum schlumbergeri, a comparatively poorly characterized species from which anatomical details are unknown. However, as previously stated, we view Saporta’s genus Cycadeospermum as having been established for ovules without histological preservation, and thus precluding assignment of the present specimens (Bateman & Hilton, 2009). The specimens described here differ from all other known fossil and extant seeds. As such, we have created a new genus and species to accommodate them, Oxfordiana motturii ART Spencer & J Hilton gen. et sp. nov. Although Saporta (1875) considered Cycadeospermum to be a cycad, this conclusion is not supported by the limited information currently available on the genus. By contrast, the structure of the integument, bilateral symmetry, and presence of an integumentary groove all suggest that Oxfordiana was a member of the Cycadales and a member of the stem-group lineage leading to the extant genus Cycas, which exhibits the same features. This conclusion agrees with the postulated Lower–Middle Jurassic divergence of Cycas from other extant genera within the Cycadales based on molecular clock studies (Nagalingum et al., 2011).

Functional morphology and herbivory

The seeds described here are larger than the majority of other recorded Jurassic seeds, an exception being some cycad seeds from the Jurassic of Haida Guai (Dawson, 1872; Chaloner & Hemsley, 1992). Within plants there is a trade-off between seed size and number: species that produce large seeds tend to do so in low numbers given a similar resource investment (Westoby, Jurado & Leishman, 1992). Larger seeds have greater independence from the environment, due to their greater resources available to support the initial stages of growth, for example under shaded conditions and periodic drought (Foster, 1986; Westoby, Jurado & Leishman, 1992; Werker, 1997). Thus, we conclude that each propagule has a greater chance of survival. We postulate that Oxfordiana seeds were probably produced in small numbers, and would have had considerable environmental independence during their early development. Furthermore, large seeds, such as Oxfordiana, are typical of non-ruderal rather than pioneering plants because they are limited to animal or water dispersal, being too large and heavy for effective transport by wind (Tiffney, 2004).

Generally, large seeds represent preferential targets for herbivory because they are more nutritious (Tiffney, 2004; Keddy, 2007). This would have been true of Oxfordiana, which nevertheless lacks obvious herbivore deterrents such as resin bodies or glands (Werker, 1997; Tiffney, 2004). However, it is possible that the integument may have contained toxins resembling those of modern cycads (Brenner, Stevenson & Twigg, 2003).

Reproductive biology

An integumentary groove as seen in Oxfordiana also occurs within the Cycadales, where it is characteristic of the extant genus Cycas in which ovules are bilaterally symmetrical (Stevenson, 1990). In Cycas this integumentary groove allows the seed to split open into two valves through the micropyle facilitating germination, a feature we interpret as also present in Oxfordiana. This situation is distinct from other extant cycads in which the embryo forces its way through the seed coat rupturing the micropylar region (Stevenson, 1990). Although we do not know the germination mechanism for Cycadeospermum (Fig. 13), we consider it probable that it too germinated through an apical groove based on its overall similarity with Oxfordiana.

The nucellar apex of Oxfordiana comprises a small pollen chamber and short nucellar beak, situated directly beneath a narrow integumentary micropylar canal. This is structurally identical to the ovules of extant Cycadales and Ginkgoales (Sporne, 1971; Singh, 1978). Post-pollination sealing of the megagametophyte in these gymnosperms is callistophytalean in nature (Serbet & Rothwell, 1995); closure occurs through both the nucellar beak and integumentary micropyle. The same form of ovule-sealing is also present within the extinct callistophytalean pteridosperms, Cordaitales and Glossopteridales, and hence differs from the hydrasperman and medullosan ovule-sealing patterns, which do not involve the post-pollination closing of integumentary tissue (Serbet & Rothwell, 1995). Cladistic analysis suggests that the callistophytalean pattern was also present in the extinct pteridosperm groups Peltaspermales, Corystospermales and Caytoniales (Rothwell & Serbet, 1994; Serbet & Rothwell, 1995). This distribution implies that it is symplesiomorphic for crown-group gymnosperms. By contrast, a coniferalean pattern is present in conifers (except Emporia), Gnetales, and the extinct Pentoxylales. Here sealing is achieved by the integument only, the pollen chamber apex remaining open post-pollination (Serbet & Rothwell, 1995). Oxfordiana lacks the central column and large pollen chamber found in hydrasperman-pattern ovules, but has the small pollen chamber and nucellar apex characteristic of the medullosan and callistophytalean patterns. We cannot determine in Oxfordiana if the integument was sealed, as in callistophytalan ovules, or open like that of medullosan ovules. Nevertheless, medullosan ovules are only known from the Palaeozoic Era and have characteristic integumentary structures lacking in Oxfordiana, we consider it more probable that Oxfordiana had post-pollination sealing that followed the callistophytalean-pattern.

Source flora, and mechanism of transportation into marine depositional setting

The source area(s) of the terrestrial fossil in the marine Oxford Clay Fm sites at Dix Pit and Marston Maisey most likely occupied the Anglo-Brabant and Welsh landmasses (based on data from Cope, Ingham & Rawson, 1992). Current palaeographic models imply a minimum of 60–70 km transport prior to deposition. Although slightly older, the Middle Jurassic Stonefield flora (Seward, 1904; Kendall, 1948; Cleal & Rees, 2003) is palaeogeographically closest to the Oxford Clay Fm sites. Cleal & Rees (2003) suggested that it represents a flora from coastal slopes surrounding the marine basin that was dominated by araucarian and cheirolepidiacean conifers, together with the probable gymnosperm Pelourdea Seward (1917). Rarer fossil plants from the site include Pinaceae (e.g. Brachyphyllum Bongniart), Bennettitales, Caytoniales, eusporangiate ferns (Dipteridaceae, Matoniaceae, Dicksoniaceae), Corystospermales, and various foliage categories resembling Cycadales and Ginkgoales (Brongniart, 1828; Cleal & Rees, 2003). The flora is thus interpreted as sampling lowland coastal habitats subjected to periodic water stress (Francis, 1983), although the Pelourdea leaves probably required fluvial transport from further inland (Cleal & Rees, 2003). Despite this, the absence from Stonefield of plants such as sphenophytes, ferns and Czekanowskiales, interpreted to have favored freshwater and wetland habits, may reflect preservation after short transportation due to the more fragile nature of the plant remains (Cleal & Rees, 2003). Le Couls et al. (2016) documented an entire cycad plant crown deposited in shallow marine sediments in the Tithonian of France that they interpreted to have inhabited coastal dunes. Numerous British lagerstätte in addition to the Oxford Clay Fm also show evidence of terrestrial plants in marine settings. These include those preserved at Bearreraig Bay (Bateman, Morton & Dower, 2000), Eathie (Wignall & Pickering, 1993), and Freeth Wood (Rothwell et al., 2007). Clearly, transportation of terrestrial plants into marine settings is widespread in the fossil record. These extremely important occurrences provide windows into past terrestrial diversity that is rarely preserved on land itself, but are limited in number (Taylor, Fueyo & Taylor, 1994; Escapa, Cúneo & Axsmith, 2008; Escapa & Leslie, 2017); none has yet been identified in the Lower Jurassic (Fig. 1B) of England or Scotland. We contend that finding additional sites for anatomically preserved plants in marine successions is important to the future development of paleobotany.

Three mechanisms of transport from the Oxfordiana plant’s growth environment to its deposition in a shallow marine sea are possible. Water transportation could occur through individual seed floatation or larger scale vegetational rafting (Carlquist, 1981; Johansen & Hytteborn, 2001), though dispersal by animal gut is also feasible (Molnar & Cifford, 2001; Zhou & Zhang, 2002). The size and weight of the ovules precludes wind transportation. We address these options individually below.

The floating capacity of seeds was investigated by Guppy (1906) and Praeger (1913), who compiled a list of seed buoyancy determined through experimentation. These studies on various floating seeds of modern coastal plants categorized three groups with contrasting floatation methods, and determined that seed size does not affect the ability to float (Guppy, 1906). Two of these groups display non-adaptive/mechanical floatation, and possess structures also seen in inland seeds (although these were generally reported to have little or no floatation power). The first group includes ovules that float because of unoccupied space in the cavity of the seed, resulting from shrinkage of the gametophyte away from the integument. In these species no other part of the structure possesses independent floatation structures (e.g. the seeds of the cycad Cycas revoluta; Giddy, 1974; Dehgan & Yuen, 1983). The second group floats either due to buoyant kernels or through layers of air-bearing tissue within the shell (e.g. the large, 2–4 cm, single-seeded eudicot Calophyllum inophyllum L.). Guppy (1906) concluded that these two categories, although able to float, have no specific adaptations for dispersal by water. In contrast, the third group represents plants that have become specifically adapted for water current dispersal. These seeds universally gain floatation power from air-bearing tissue in their seed-coats, for instance the extra “spongy” parenchymatous tissue of the cycadalean Cycas circinalis L., which can float for between five weeks and three months (Ridley, 1930; Gunn & Dennis, 1976; Dehgan & Yuen, 1983).

Assuming that the Oxford Clay Fm ovules represent the full range of tissues present in life, there are options for placing the species within Guppy’s floatation categories. They cannot be placed in the third group because no seed-coat air-pocket tissues were present in the Oxford Clay Fm ovules—rather, their tissue layers are made of densely packed cells. This lack of floatation tissue also rules out the second group. Hence, the Oxford Clay Fm ovules must belong to the first group—seeds that have a large air-tight internal cavity, as seen in many modern cycad species lacking specialized floatation tissue. In the fossil ovules the gametophytic tissue was not fully expanded forming an air-cavity. In some cycad species low gametophytic growth and the corresponding ability to float has been linked to seed viability—in which fertilized seeds tend to sink (Dehgan & Yuen, 1983). If a similar mechanism applied to the Oxford Clay Fm ovules, it would imply that they were at a pre-fertilization stage.

Another method of transport from source to depositional site is vegetational rafting. This form of chance dispersal relies on seeds being carried out to sea by rafts of floating plant debris (Carlquist, 1981). Little work has been done to study seed-deposition through vegetational rafting; the emphasis of most published studies is on the trans-ocean transport of fauna. However, one study has demonstrated that some modern-day coastal plants use rafting to transport their seeds in the North Atlantic/Arctic Ocean (Johansen & Hytteborn, 2001). Today, rafts form regularly at higher latitudes through river transportation from large amounts of floating wood and entangled vegetation. This occurs with decreasing regularity at lower latitudes due to increased inter-annual variation in the availability of wood (Gibson, Atkinson & Gordon, 2006). We note that rafting is a possibility here: the fossil wood found within the Oxford Clay Fm, occurring as logs up to 1 m long and 80 cm in diameter (Porter, 1863). Hence, the material for raft building existed, and was circulating in the basin waters prior to deposition.

During the Jurassic, large land-dwelling herbivores, such as prosauropod (Weigelt, 1930) and sauropod (Stokes, 1964; Mohabey, 2001) dinosaurs, became the dominant terrestrial herbivores (Tiffney, 2004). The dentition of these suggests they relied on gut gastroliths coupled with gut fermentation to break down the plant material ingested (Farlow, 1987). This style of digestion probably allowed whole seeds to enter the gut (Tiffney, 2004; Mustoe, 2007). That these animals lived on land reduces the possibility that they were the transport vector for the ovules while alive. In death, however, they evidently reached the marine Oxford Clay Fm depositional zone, as the unit contains abundant body fossils (see Martill & Hudson, 1991, and references therein). Therefore, the possibility—albeit a very remote one—exists that during the decomposition of a dinosaur at sea, undigested plant matter was released into the sea. Pterosaurs were present in the marine environment. Although study of their dentition suggests the group were primarily suited for piscivory or insectivory, they have been implicated as potentially important seed dispersers by Fleming & Lips (1991). The piscivorous pterosaur Rhamphorhynchus is known from the Oxford Clay Fm (Martill et al., 1994), but frugivores, such as Tapejara (Wellnhofer, 1991; Wang & Zhou, 2003), are not currently recognized before the Cretaceous. The oldest bird-like fossil, Aurornis, from the Jurassic Callovian–Kimmeridgian stages of China (Godefroit et al., 2013), is coeval with the Oxford Clay Fm ovules; however, doubt has been cast on the authenticity of this early date (Balter, 2013). The next oldest fossil avian, Archaeopteryx, is Tithonian in age (so significantly postdates deposition of the Oxford Clay Fm) and is interpreted to have been carnivorous (Ostrom, 1976). However, seed-eating birds are known from the Lower Cretaceous of China (Zhou & Zhang, 2002; Zhou, 2004). They consumed small, circular (8–10 mm long) seeds of the Carpolithus-type (Zhou & Zhang, 2002); these are very different from the large seeds of Oxfordiana. Hence, we argue that dispersal of large seeds, such as Oxfordiana, into marine settings by winged vertebrates is, based on current evidence, unlikely in the Jurassic.

Mode of preservation

The specimens described here have different modes of preservation. In the case of BU 5265, it appears that rapid burial limited bacterial decomposition, thus preserving the ovule in an almost perfect state. The lack of taphonomic degradation during transportation to the depositional site is, however, intriguing. If the ovule floated via a self-contained mechanism, until becoming inundated and sinking (see above), the effects of salt water and bacterial degradation were minimal. It is possible, if this scenario is true, that the ovule seed-coat may have evolved to be both waterproof (i.e. a impermeable/hard seed; Barton, 1967) and to have possessed a degree of antibacterial properties (e.g. high tannin levels; Roth, 1987). Fossil data are not able to add any support to this conjecture; no cells have been found that could be defined as wax- or oil-producing for aiding waterproofing, nor can the chemical properties of the seed-coat be deduced. If transportation occurred through rafting, the ovule could have remained relatively dry until the point of inundation (when the raft breaks up or sinks). Assuming that the seed did not possess any adaptations for hydrochory, this phenomenon could have extended the time before taphonomic degradation became a significant factor, and also allowed the seed to sink rapidly to the sea floor for burial.

In contrast, specimen BU 5266 has more extensive pyritization, including disruptive pyrite growth in the plant tissues (Figs. 4, 7A and 7B), suggesting a greater period of exposure to salt water prior to burial and permineralization. Water saturation of the ovule would have resulted in voids filling with sea-water. If transport occurred through floatation, this would have led to sinking and burial. If rafting was involved, the ovule may have remained on the sea surface whilst fully saturated—ideal conditions for bacterial growth (i.e. warm, high oxygen, moist).

After sinking and burial in seafloor sediments, both ovules underwent preservation in two distinct mineralization phases: first rapid pyritization, followed by a second carbonate phase. The latest Callovian Stage to early Oxfordian Stage seafloor is interpreted to have been tens of meters deep in this area (Martill & Hudson, 1991), and has been described as having a dominantly firm substrate. Reports suggest that pockets of “soupy” sediment existed, within which rapid burial of organic specimens could take place (Martill & Hudson, 1991; Wilby et al., 2004). Previous studies of taphonomic processes and pyrite preservation in the Oxford Clay Fm have focused on the fauna (e.g. ammonites, belemnites) within these organic-rich sediments (Martill & Hudson, 1991).

The process of pyritization within the Oxford Clay Fm commenced through anoxia of the host sediment, which causes sulfate reduction of carbon within the sediment (6–16%; Kenig et al., 1994; Raiswell, 1997). The sulfate produced reacts with iron in the sediment to yield iron monosulfate (FeS). This subsequently reacts with hydrogen sulfide created by sulfate-reducing bacteria decomposing sedimentary organic matter (Rickard & Luther, 1997). A by-product of this process is increased water alkalinity (Berner, 1970, 1984; Berner et al., 1985; Skyring, 1987). Iron monosulfate porewater concentration increases, almost reaching saturation point—a process seen in modern continental margin sediments (Raiswell, 1997). Precipitation of iron and sulfide minerals, such as mackinawite, or direct pyrite precipitation results. For the Oxford Clay Fm ovules this represents the first phase of mineralization, responsible for depositing pyrite within and around cell walls. These cells would have been undergoing bacterial decay by sulfide reduction, while the ovule was becoming buried in successive surface layers of the sediment (Fig. 14).

Figure 14 Fossilization phase diagram.

Illustrating the qualitative changes at varying depths/zones (A) in dissolved iron and sulfide (B) and how these changes affect the saturation state between iron and carbonate during microbial diagenesis (C), as well as showing the related different styles of fossilization (D). (E) an SRXMT longitudinal section of BU 5265.1 showing the two phases of mineralization. See Figs. S1A–S1D for more SRXMT section images. Phase data based on Raiswell (1997).

At the same time as pyrite mineralization was occurring, due to the lower saturation of carbonate within the porewater system, contemporaneously deposited shelly fauna (including ammonites and belemnites) were undergoing carbonate shell dissolution and replacement by pyrite. Carbonate dissolution and pyrite replacement would have occurred until such time as the carbonate concentration increased, allowing direct precipitation of pyrite on to the shells as a coating (Raiswell, 1997). As the Oxford Clay Fm ovules became buried at greater depth (1–10 m), over-saturation of iron decreased, reducing the ability of pyrite to form and at the same time the carbonate saturation increased to become over-saturated. Under a methanogenesis regime in this over-saturated state, deposition of carbonate within the ovule void spaces occurred, while the pyrite deposition decreased and ultimately terminated (Fig. 14).

Preservation of the ovules thus occurred through floating/rafting, then sinking into soft sediment, and subsequent pyritization with secondary carbonate infilling. This series of events is unlikely to have occurred regularly, and is one possible reason for the rarity of fossil seeds reported from the Oxford Clay Fm despite its long history of fossil collecting (Martill & Hudson, 1991). We note also that the majority of historic collecting may have been focused on the fauna rather than the flora, leading to a Kingdom-level collecting bias. Preservation of these seeds is different from fossil woods preserved elsewhere in the deposit where pyrite growth tends to be amorphous, and rarely preserves cellular features. We suggest that these specimens concentrated sulfate-reducing bacteria into a closed microenvironment that allowed exceptional preservation. By contrast, larger specimens, such as plant trunks and branches, would have had more diffuse mineralization at lower rates over their extent. They probably also lacked the sealed microenvironments required for exceptional preservation (Grimes et al., 2001, 2002; Tibbs, Briggs & Prössl, 2003). Finally, we would emphasize that pyritization does not preserve cell contents. Rather it envelops cell walls and tissue layers: an aggrading process that can make cell size measurements using traditional optics difficult and may lead to exaggerated estimates. Using SRXMT has here allowed us to overcome this taphonomic effect by measuring the carbonate preserved cell walls embedded within the pyrite coatings (Figs. 6, 7 and 9).

Conclusions

Oxfordiana motturii gen. et sp. nov. is a distinctive gymnospermous ovule that differs from all other fossil and extant seeds, and is here placed within the Cycadales as a member of an extinct, stem-group lineage relative to the extant genus Cycas. We conclude that it had valvate germination through a distal integumentary groove, as well as a callistophytalean pattern of ovule-sealing. We also demonstrate that SRXMT can provide valuable new insights into sectioned fossil specimens and, in particular, their taphonomic history.

We thank the PeerJ editor R.M. Bateman for the constructive comments and suggestions that have helped in the revision of this manuscript. Additionally, the text was improved by insightful reviews from C.J. Cleal and an anonymous reviewer whom we thank. We thank Michael Drakopoulos, Nghia Vo, Christina Reinhard, Robert Atwood and Kaz Wanelik for their support and assistance during this beamtime. We thank Alice Spencer for help with Latin translations. This work constituted part of a doctoral research project for ARTS at Imperial College London for which Mark Sutton and members of the Earth Science and Engineering Department are thanked.

Additional Information and Declarations

Competing Interests

Author Contributions

Data Availability

New Species Registration

The authors declare that they have no competing interests.

Alan R.T. Spencer conceived and designed the experiments, performed the experiments, analyzed the data, contributed reagents/materials/analysis tools, wrote the paper, prepared figures and/or tables.

Russell J. Garwood conceived and designed the experiments, performed the experiments, reviewed drafts of the paper.

Andrew R. Rees reviewed drafts of the paper.

Robert J. Raine contributed reagents/materials/analysis tools, reviewed drafts of the paper.

Gar W. Rothwell analyzed the data, reviewed drafts of the paper.

Neville T.J. Hollingworth reviewed drafts of the paper.

Jason Hilton conceived and designed the experiments, performed the experiments, analyzed the data, contributed reagents/materials/analysis tools, wrote the paper, reviewed drafts of the paper.

The following information was supplied regarding data availability:

Zenodo–https://zenodo.org

Video S1: https://doi.org/10.5281/zenodo.61841

Video S2: https://doi.org/10.5281/zenodo.61841

Video S3: https://doi.org/10.5281/zenodo.61841

Model S1: https://doi.org/10.5281/zenodo.61841

Model S2: https://doi.org/10.5281/zenodo.61841

Figure S1: https://doi.org/10.5281/zenodo.61841

Dataset S1: https://doi.org/10.5281/zenodo.824099

Dataset S2: https://doi.org/10.5281/zenodo.824103

Dataset S3: https://doi.org/10.5281/zenodo.824029

Dataset S4: https://doi.org/10.5281/zenodo.824047

Dataset S5: https://doi.org/10.5281/zenodo.824051

Dataset S6: https://doi.org/10.5281/zenodo.824073

Dataset S7: https://doi.org/10.5281/zenodo.824079

Dataset S8: https://doi.org/10.5281/zenodo.824085

Dataset S9: https://doi.org/10.5281/zenodo.824089

Dataset S10: https://doi.org/10.5281/zenodo.824091.

The following information was supplied regarding the registration of a newly described species:

Genus name: Oxfordiana

Species name: motturii

Note: Fossil taxa

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
