# Peer review of "New insights into Mesozoic cycad evolution: an exploration of anatomically preserved Cycadaceae seeds from the Jurassic Oxford Clay biota"

_PeerJ, doi:10.7717/peerj.3723_

## Round 0.1 · original submission · Minor Revisions

I selected two reviewers who are well-qualified to review this manuscript yet have failed to acquire fearsome reputations as harsh reviewers, and so it has proved in this case (indeed, Reviewer 2 has proven so shy as to opt for anonymity!). Effectively, the reviewers have elected to copy edit your submission rather than bore through its crust to explore its soft-science mantle and hard-science core.

I'm afraid that, having just returned from a major field trip, I too am obliged to adopt this suboptimal approach. Happily, the reviewers and I appear to concur that this is an unusually thorough study utilising modern techniques and leading to a discussion that (at least, in the context of palaeobotanical research) provides an appropriate breadth of avenues speculatively explored. Both reviewers note that the text could be shortened somewhat, but I see no value in doing so – Armstrong and Aldrin could in theory have cut short their trip to the Sea of Tranquillity and instead landed on the beach at Clacton, but I doubt that their arrival would have achieved quite the same historical impact. Most of the reviewer's comments can easily be accommodated (e.g. by stating that only five UK localities have yielded intact organs showing anatomical preservation). I would add my weight to reviewer 2, who rejects the radial symmetry of the overall ovules in favour of 180 degree rotational symmetry – indeed, this conclusion is dictated by the linear apical ridge. Perhaps ovules are better treated like angiosperm flowers? – you need to describe the symmetry of each layer of tissue rather than the organ (in this case, ovule) as a whole.

Although it is fairly well-prepared, I would encourage the authors to work through the manuscript one more time. As far as I can see, there are plenty more errors of spelling, punctuation, English usage and journal formatting than those few picked up en passant by the two reviewers. For example, they raise the issues of chrono- and lithostratigraphic nomenclature, but not preferred units of measurement of time (I prefer the USGS's Ma for absolute time and Myr for relative time… cf. line 176). I thought that I had weaned the author's off such meaningless phrases as "a number of" (line 319), and it might be helpful to add the missing z to Czekanowskiales (line 501). In the descriptions, an x is used where there should a multiplication symbol, and it lacks the spaces needed to top and tail the symbol (e.g. line 356). I could go on (and on).

My one more substantive comment is that the author's chosen title initially suggested to me a manuscript of such unremitting tedium that reading it might cause me to involuntarily gnaw off my own leg! Surely the technology used here with a degree of innovation, and/or the suggestion of a substantially new category of a seed plant, warrant inclusion?

But ultimately, I leave these weighty decisions for the author's adjudication, and look forward to seeing this paper published with alacrity.

·

Basic reporting

The paper is on the whole well-written, illustrated and structured, and I have few comments. There are, however, a few minor grammatical mistakes (?typos).
Line 29 should read "represents" instead of "represents to"
Line 34 should read "fossils" instead of "fossil"
Line 126 should read "models that were" instead of "models were"
Line 164 - delete "unit of" at end of line
Line 205 should read "that" instead of "suggesting"
Line 208 - delete "organised" at end of line
Line 220 should read "organised in" instead of just "organised"
Line 264 - delete comma after "into"
Line 321 - delete colon after "which"
Line 373 should read "are a group" instead of "are group"
Line 425 should read "from" instead of "form"
Line 443 - delete "all"
Lines 445-447 - sentence needs rewording
Line 476 - "Caytonia" should be italicised
Line 501 should read "sphenophytes" instead of "sphnophytes" - and this sentence as a whole needs re-wording
Line 505 should read "to have inhabited" instead of "to inhabit"
Line 527 - I would prefer "because of" rather than "through" at the end of the line (it sounds as though the ovule is floating through unoccupied space!)
Lines 540-542 - sentence needs re-wording (maybe just deleting "as" on 541)
Lines 544-545 - again needs re-wording.

Capitalisation of certain "informal" taxonomic names (e.g. "Cycads") should be looked at - only formal names should really be capitalised.

Some regard it as pedantic, but chronostratigraphical terms should be adjectival, not nouns. For instance, line 175 should read "Callovian Stage" not just "Callovian"; line 345 should read "the Palaeozoic Era" or "Palaeozoic times", not just "the Palaeozoic". And there is no such thing (line 301) as the "uppermost Callovian to basal Oxfordian Stages" - it should read "uppermost Callovian to basal Oxfordian stages".

References. In the literature review (Introduction) no mention is made of the Purbeck fossils such as described by Seward (1897, QJGS, 53, 22-39) but these may just be earliest Cretaceous in age - I'll let the authors check that one!

Experimental design

The paper certainly fits within the scope of the journal. It describes new fossils representing (probably) a new species, and tries to determine their systematic position. Nothing really comparable is known from the British Isles so in that sense alone it fills a knowledge gap. It is based on some of the most up-to-date methods - I cannot comment on the adequacy of the methods description, but to my "un-tutored eye" they look OK.

Validity of the findings

The data are (not "is"!!!!) robust and their conclusion as to the systematic position of the fossils is probably as far as it can be taken. The authors keep on talking about "pteridosperms" as though they represent a meaningful group - I wish I knew what that group was! And on lines 379-380 it is implied that the Caytoniales and Pentoxylales were "pteridosperms" - whatever is being meant by "pteridosperms", I am sure these two orders wouldn't be included!

Additional comments

Basically the paper is good - most of the above comments are minor

Reviewer 2 ·

Basic reporting

The text is generally clear, but there is room for some grammatical improvement and clarifications of ideas.

The authors should follow the journal’s style VERY carefully. Use a paper from the most recent issue as a guide to style. Be especially careful with the formats for headings, spacings, paragraph indentations, the use of en dashes versus hyphens versus em-dashes, and reference formats.

The layout of the manuscript is appropriate, although some parts might be trimmed a little.

Be careful with the use of chronostratigraphic vs geochronologic units. Upper and Lower refer to rock packages and their positions in the stratigraphic column; early and late refer to time packages.

Experimental design

This manuscript describes a new genus and species based on permineralized seeds from the Oxford Clay at two sites in the UK.

In general, the seeds are not particularly well preserved compared to some silicified and calcified forms from sinter and permineralized peat deposits. However, this is one of those manuscripts that shows what can be achieved by subjecting the fossils to a battery of destructive and non-destructive analytical techniques accompanied by tomographic reconstructions.

The methods are adequately described to enable repetition of such a study.

Validity of the findings

The authors have uncovered some interesting Middle to Late Jurassic seeds that can not be ascribed readily to any of the typical mid-Mesozoic plant groups. The size, thick integument and radial (or perhaps 180° rotational) symmetry of the seeds shows greatest similarities with Cycadales but suggests that the true diversity of seed plants in the mid-Mesozoic was much greater than we have gleaned from leaf impression assemblages of delta-plain mires.

The interpretations are reasonable, and the illustrations adequately document the anatomy of the various components of the seeds. This paper goes further than most in that it addresses the probable post-depositional diagenetic pathway that led to the curious mode of permineralization of these seeds in a marine setting.

Additional comments

There are several minor issues that need addressing before publication. The official name of the host unit (and of other units mentioned in the text and figures) needs to be clarified. The title uses the term ‘Oxford Clay’ for the host unit. Elsewhere, the term ‘Oxford Clay Formation’ is used. The International Stratigraphic Guide recommends that formation names be given a single geographic identifier and a lithological modifier. In this case Oxford Clay (or Claystone) would be standard. However, I am aware that for historical reasons some unit names have diverged from this approach and have a double modifier (Oxford Clay Formation). Surely one of these must be the ‘official’ name according to the relevant national stratigraphic nomenclature committee. Whatever this is, it should be used consistently throughout.

Line 62-63. I am surprised that there are only five ‘sites’ yielding permineralized plants in the UK Jurassic. But does this list include all records of permineralized wood? – I would have thought that wood had a much wider geographic and greater stratigraphic representation.

According to the International Stratigraphic Guide, formally defined biostratigraphic units should be written in full (or with the genus name abbreviated) and the unit should be in capitals; e.g., the Q. lamberti Zone. They should not be reduced to just lamberti Zone (although many lazy authors have done this in the past).

If space is at a premium, some sections could be trimmed. For example, comparisons with Leptostrobales, which had diminutive seeds could be excluded. Some of the discussion sections dealing with taphonomy and functional biology could also be trimmed as they begin to venture into speculative territory with regards to modes of transport into the marine environment.

I have not checked whether all references are present in the list. Formatting should be checked for consistency with the journal.

The illustrations are of good quality given the nature of the material and adequately document the range of analytical techniques used and the anatomical features encountered in the seeds.

Additional comments are indicated on the attached annotated pdf.

---

## Round 0.2 · Minor Revisions

A swift change of title, as nobly agreed five minutes ago, and we can move this manuscript through to publication pronto ...

---

## Round 0.3 · accepted · Accept

Thank you – a really nice paper put safely to bed (I hope). Though I may need to reconsider my decision if you decide to send the silly tweet suggested by the journal ...